# Cytotoxic Polyketide Metabolites from a Marine Mesophotic Zone Chalinidae Sponge-Associated Fungus *Pleosporales* sp. NBUF144

**DOI:** 10.3390/md19040186

**Published:** 2021-03-26

**Authors:** Jing Zhou, Hairong Zhang, Jing Ye, Xingxin Wu, Weiyi Wang, Houwen Lin, Xiaojun Yan, J. Enrico H. Lazaro, Tingting Wang, C. Benjamin Naman, Shan He

**Affiliations:** 1Department of Marine Pharmacy, Li Dak Sum Marine Biopharmaceutical Research Center, College of Food and Pharmaceutical Sciences, Ningbo University, Ningbo, Zhejiang 315800, China; 13567427589@189.cn (J.Z.); heazin@163.com (H.Z.); yanxiaojun@nbu.edu.cn (X.Y.); bnaman@nbu.edu.cn (C.B.N.); 2State Key Laboratory of Pharmaceutical Biotechnology, School of Life Sciences, Nanjing University, Nanjing 210023, China; DG1930079@smail.nju.edu.cn (J.Y.); xingxin.wu@nju.edu.cn (X.W.); 3Key Laboratory of Marine Biogenetic Resources, Third Institute of Oceanography, Ministry of Natural Resources, Xiamen 361005, China; wywang@tio.org.cn; 4State Key Laboratory of Oncogene and Related Genes, Department of Pharmacy, Research Center for Marine Drugs, Ren Ji Hospital, School of Medicine, Shanghai Jiao Tong University, Shanghai 200127, China; franklin67@126.com; 5National Institute of Molecular Biology and Biotechnology, University of the Philippines Diliman, Quezon 1101, Philippines; jaylazaro@mbb.upd.edu.ph

**Keywords:** sponges, fungi, sponge-associated fungi, polyketide, cytotoxicity, mesophotic coral ecosystems, twilight zone

## Abstract

Two new polyketide natural products, globosuxanthone F (**1**), and 2′-hydroxy bisdechlorogeodin (**2**), were isolated from the fungus *Pleosporales* sp. NBUF144, which was derived from a 62 m deep Chalinidae family sponge together with four known metabolites, 3,4-dihydroglobosuxanthone A (**3**), 8-hydroxy-3-methylxanthone-1-carboxylate (**4**), crosphaeropsone C (**5**), and 4-megastigmen-3,9-dione (**6**). The structures of these compounds were elucidated on the basis of extensive spectroscopic analysis, including 1D and 2D NMR and high-resolution electrospray ionization mass spectra (HRESIMS) data. The absolute configuration of **1** was further established by single-crystal X-ray diffraction studies. Compounds **1**–**5** were evaluated for cytotoxicity towards CCRF-CEM human acute lymphatic leukemia cells, and it was found that **1** had an IC_50_ value of 0.46 µM.

## 1. Introduction

Marine organisms, especially sponges and microbes associated with them, are prolific producers of structurally diverse and bioactive natural products [1,2,3,4,5]. As sessile filter-feeding animals, sponges provide habitats for diverse and specific microbial communities [6]. The term “holobiont” is used to describe multilevel assemblages of organisms composed, for example, of a sponge and associated microbiota (up to 40% of sponge volume), including fungi, yeast, and bacteria [7,8,9,10]. Sponge-associated fungi have produced many intriguing molecules reported with conspicuous pharmacological and biological properties, exemplified by antimicrobial [11,12,13], cytotoxic [13,14], anti-viral [15,16], antifungal [17], anti-inflammatory [18], and α-glucosidase inhibitory [19] activities. However, a vast quantity of sponge-derived fungal biodiversity remains uninvestigated, suggesting that many natural product metabolites also remain to be discovered.

Mesophotic coral ecosystems (MCEs or “Twilight zone” coral reefs) range from 30 to 150 m deep and represent approximately 80% of potential coral reef habitats worldwide, but little is known about them compared with shallow reefs [20,21]. In an ongoing investigation of fungal metabolites [22,23], rare or unexplored sponge-derived fungi from MCEs have attracted the attention of these authors and others. Several fungi with novel metabolite-producing potential were prioritized for further study after applying the combined strategy of LC-MS/MS molecular networking [24] with the OSMAC (One Strain MAny Compounds) approach [25] of crude extracts from 80 sponge-associated fungi (40 from shallow water and 40 from MCEs; Appendix A). For example, in the previous work, the organism first prioritized was a *Cymostachys* fungus that yielded a series of new polyketide compounds, cymopolyphenols A-F [26]. Among them, cymopolyphenols D-F are low-order polymers of cymopolyphenols A that have novel chemical scaffolds. As a continuation of this work of mining new or novel fungal metabolites from MCEs, a second organism was prioritized from the same strain library after the previously reported molecular network [26] was adapted to highlight the secondary metabolite-producing activity of a *Pleosporales* sp. NBUF144 fungus, which was derived from a 62 m deep Chalinidae family sponge (Appendix A). The chemical investigation of this organism yielded compounds **1**–**6** (Figure 1)**,** and the details of these experiments and associated biological testing results are reported herein.

## 2. Results and Discussion

### 2.1. Structure Elucidation and Identification of Compounds

Compound **1** was determined to have the molecular formula of C_15_H_12_O_7_, based on a sodiated molecular ion peak at *m/z* 327.0481 [M + Na]^+^ (calcd for C_15_H_12_O_7_Na, 327.0475) in the high-resolution electrospray ionization mass spectra (HRESIMS), indicating 10 degrees of unsaturation. The ^1^H and ^13^C NMR spectra of **1** indicated the presence of one methoxy (*δ*_H_ 3.92, *δ*_C_ 54.4; 10-O*CH*_3_), three hydroxy (*δ*_H_ 4.42; 1-OH, *δ*_H_ 2.93; 2-OH, *δ*_H_ 12.13; 8-OH), five protonated olefins (*δ*_H_ 6.50, *δ*_C_ 143.8; CH-3, *δ*_H_ 6.32, *δ*_C_ 120.5; CH-4, *δ*_H_ 6.90, *δ*_C_ 107.3; CH-5, *δ*_H_ 7.52, *δ*_C_ 135.8; CH-6, *δ*_H_ 6.80, *δ*_C_ 112.1; CH-7), two carbonyls (*δ*_C_ 180.7; C-9, *δ*_C_ 174.7; C-10), and non-protonated carbons (*δ*_C_ 73.2; C-1, *δ*_C_ 160.9; C-4a, *δ*_C_ 160.8; C-8, *δ*_C_ 111.0; C-8a, *δ*_C_ 180.7; C-9, *δ*_C_ 113.7; C-9a; *δ*_C_ 155.5; C-10a) (Table 1). Cross peaks detected in the ^1^H-^1^H COSY spectrum led to the generation of two isolated spin systems, H-5−H-6−H-7 and 2-OH−H-2−H-3−H-4. HMBC correlations from H-5/H-7 to C-8a, H-6 to C-10a/C-8, and 8-OH to C-8a/C-7 revealed one *o*,*m*-disubstituted phenolic group (Figure 2). Further HMBC signals observed from H-3 to C-1/C-4a, H-4 to C-9a, 1-OH to C-2/C-9a/C-10, and 10-O*CH*_3_ to C-10 in HMBC established a methyl 1,6-dihydroxy-2,4-cyclohexadiene-1-carboxylate moiety (Figure 2). The co-existence of one unassigned carbonyl (*δ*_C_ 180.7; C-9) and the deshielded shifts of two non-protonated sp^2^ carbons that suggested linkage to an oxygen atom (C-4a at *δ*_C_ 160.9 and C-10a at *δ*_C_ 155.5) indicated the presence of a pyran-4-one moiety connecting the two identified spin systems (Figure 2), which together satisfied the remaining requirement of two degrees of unsaturation. The methoxy group had an HMBC correlation with C-10, conclusively assigning its position of attachment to the scaffold. Therefore, the planar structure of **1** was determined to be methyl 1,2,8-trihydroxy-9-oxo-2,9-dihydro-1H-xanthene-1-carboxylate, which matched that of globosuxanthone A [27,28,29]. Fortunately, compound **1** was able to be crystalized in a mixed organic solvents system (MeOH/DCM/H_2_O, 40:20:1), and it was further studied by single-crystal X-ray diffraction. This allowed the (1*S*,2*R*) configuration to be assigned to **1** (Figure 3) with a Flack parameter of −0.13(12) [30]. The 1-OH and 2-OH groups in **1** were determined to be *cis* oriented, while the corresponding groups in globosuxanthone A are *trans*. Therefore, compound **1** was assigned as a new natural product representing the C-1 epimer of globosuxanthone A, here named globosuxanthone F.

Compound **2** was afforded as colorless prisms. The molecular formula was determined to be C_17_H_14_O_8_ based on the sodium adduct peak at *m*/*z* 369.0587 [M + Na]^+^ (calcd. for C_17_H_14_O_8_Na, 369.0581) in the HRESIMS, requiring 11 degrees of unsaturation. The ^1^H and ^13^C NMR spectra of **2** revealed the occurrence of one methyl (*δ*_H_ 2.39, *δ*_C_ 23.2; C-6′), two methoxy (*δ*_H_ 3.44, *δ*_C_ 51.4; C-1″-*O*-CH_3_, *δ*_H_ 3.69, *δ*_C_ 57.1; C-5′-O*CH*_3_), three olefins (*δ*_H_ 6.40, *δ*_C_ 104.6; CH-5, *δ*_H_ 6.35, *δ*_C_ 109.0; CH-7, *δ*_H_ 5.82, *δ*_C_ 102.5; CH-4′), eight non-protonated carbons (*δ*_C_ 93.4; C-2, *δ*_C_ 108.2; C-3a, *δ*_C_ 171.8; C-4, *δ*_C_ 152.7; C-6, *δ*_C_ 155.8; C-7a, *δ*_C_ 149.6; C-1′, *δ*_C_ 148.3; C-2′, *δ*_C_ 171.2; C-5′), one unconjugated and one conjugated carbonyl (*δ*_C_ 198.1; C-3, *δ*_C_ 180.9; C-3′), and one ester (*δ*_C_ 167.7; C-1”). HMBC correlations were observed from H-5 to C-3a, H-7 to C-5/3a, and H-6′-CH_3_ to C-5/C-7 that established an *m*-cresol moiety (Figure 2). Further correlations from H-1”-*O*-CH_3_ to C-1”, H-4′ to C-2′/C-3′/C-5′, and H-5′-*O*-CH_3_ to C-5′ in HMBC suggested the hallmarks of methyl 2-hydroxy-5-methoxy-3-oxocyclohexa-1,4-diene-1-carboxylate motif as is present in the known fungal metabolite bisdechlorogeodin [31]. From this substructure, one carbonyl and an oxygen atom remained unassigned, together with two double bond equivalents. However, this was determined to indicate a furan-3(2*H*)-one moiety in **2** as shown in Figure 1 rather than pyran-4-one motifs as in **1**, because only the deshielded quaternary carbon C-2 could be assigned to both this central ring system and the cyclohexadienone moiety. A molecule with the same carbon skeleton as **2** was previously reported as bisdechlorogeodin [31]. However, comparison of the NMR data of **2** with that of bisdechlorogeodin highlighted that an olefinic proton at *δ*_H/C-__2′_ 7.11/137.0 in bisdechlorogeodin was absent in **2**, and instead an oxygenated sp^2^ carbon was observed at *δ*_C_ 148.3 in **2**, reflecting the 2′-hydroxy group in this new molecule. Single-crystal X-ray diffraction analysis was attempted to establish the C-2 configuration of **2** (Figure 3), but since a centrosymmetric space group (*P*_-1_) was obtained with the crystals, only confirmation of the planar structure of **2** was achieved. However, on the basis of biosynthetic logic, it was suggested that **2** should share the 2*R* configuration of bisdechlorogeodin [31]. Since the additional 2′-OH group in **2** compared to bisdechlorogeodin is understood to impact less the optical rotation (OR) of the molecule than inverted configuration would, the absolute configuration of C-2 in **2** was assigned as *R* based on the comparable OR data of (−)-bisdechlorogeodin{[*α*]d-107 (*c* = 0.1, in EtOH) [32]; [*α*]d-154 ± 12} [33] and compound **2** {[*α*]^25^d-160 (*c* = 0.1, in MeOH)}. Furthermore, the crystal structure observed for **2** leaves no chance that the additional hydroxy group in this compound would participate in intra-molecular hydrogen bonding that could lead to an ambiguous OR phenomenon that was recognized recently in some analogs of (+)-egenine [34]. Accordingly, the new natural product analog of bisdechlorogeodin, **2**, was given the trivial name 2′-hydroxy bisdechlorogeodin.

Compounds **3**–**6** were identified as 3,4-dihydroglobosuxanthone A (**3**) [35], 8-hydroxy-3-methylxanthone-1-carboxylate (**4**) [36], microsphaeropsone C (**5**) [35], and 4-megastigmen-3,9-dione (**6**) [37] by comparison of obtained experimental NMR, MS, and OR data with literature values.

### 2.2. Bioactivity Assay

Globosuxanthone A has been reported to yield significant cytotoxicity towards eight human solid tumor cell lines, including NCI-H460, MCF-7, SF-268, PC-3, PC-3M, LNCaP, DU-145, and HCT-15, as well as T-cell leukemia Jurkat cells [27,29]. These data inspired in vitro cytotoxicity testing of the compounds isolated here. Compound **1** exhibited potent cytotoxicity in vitro against CCRF-CEM T-cell leukemia cells with a IC_50_ value of 0.46 μM. Compounds **3**–**5** each showed no pronounced cytotoxicity at 20 μM (Figure 4), despite sharing almost identical scaffolds with globosuxanthone A and **1**, suggesting the importance of the Δ3,4 unsaturation (comparing **1** to **3**) and the presence of hydroxy groups at C-1 and C-2 in this scaffold (comparing **1** to **4**) for yielding cancer cell cytotoxicity. Compound **6** was not obtained in sufficient quantity for testing in this study.

## 3. Materials and Methods

### 3.1. General

Optical rotations were measured with a JASCO P-2000 automatic polarimeter in MeOH at 20 °C. The CD spectra were recorded on a JASCO P-1500 spectropolarimeter. NMR spectra were recorded in CDCl_3_ using residual solvents as internal standards with a Bruker AVANCE NEO 600 spectrometer with a 5 mm inverse detection triple resonance (H-C/N/D) cryoprobe having z-gradients. The chemical shift values are given in parts per million (ppm) relative to TMS at 0.0 ppm, and the coupling constants are in Hertz. High-resolution electrospray ionization mass spectra (HRESIMS) were measured on an Agilent (Santa Clara, CA, USA) 6545 Q-TOF instrument. Reversed-phase HPLC purification was carried out on a Waters HPLC equipped with a 1525 binary pump, and a Thermo Scientific (Waltham, MA, USA) ODS-2 Hypersil column (5 µm, 250 × 10 mm). Normal phase column chromatography and thin-layer chromatography were performed using silica gel (200–300 mesh) and GF254 (10–20 mm) (Qingdao Marine Chemical Company, Qingdao, China). YMC*GEL ODS-A (AA12S50; YMC Co., Ltd., Japan) was used for reverse phase column chromatography. Sephadex LH-20 was a product from GE Biotechnology, USA.

### 3.2. Fungal Strains

The fungal strain was isolated from a Chalinidae family sponge collected in 2018 by scientific-technical SCUBA diving at the depth of 62 m near Apo Island, Negros Oriental, Philippines (9°04′40.6″ N 123°15′57.3″ E and 9°04′33.0″ N 123°15′59.1″ E). The inner tissue of the sponge was sliced into squares (0.5 × 0.5 × 0.5 cm^3^) and incubated on plates containing modified fungal mediums described elsewhere [22]. The sponge sample loaded petri-dishes were then incubated at 28 °C for two weeks. The fungal colonies were picked and sub-cultivated on PDA (Potato Dextrose Agar; potato 200.0 g, glucose 20.0 g, sea salt 35.0 g, agar powder 20.0 g, and H_2_O up to a total volume of 1 L) media.

The fungal strain further grown and evaluated in this study was identified as a *Pleosporales* sp. based on morphological traits and sequence analysis of the ITS region (GenBank accession no. MW386403).

### 3.3. Cultivation of the Fungi

For chemical investigation, the *Pleosporales* sp. was cultured in 250 × 1 L Erlenmeyer flasks, each containing 400 mL PDB medium (80 g potato, 8 g glucose, 14.0 g sea salt, 400 mL H_2_O), for 15 days at 28 °C with agitation (120 rpm). The culture broth was filtered through cheesecloth to separate it into filtrate and mycelia (retained in the cheese), and both of them were extracted with EtOAc (*v*/*v*, 1:1) for 5 times and MeOH and DCM (*v*/*v*, 1:1) for 3 times, respectively. The separated extracts of the filtrate and mycelia were combined for their similar spots in TLC analysis to afford a total crude organic extract (280 g).

### 3.4. Extraction and Isolation

The crude extract was subjected to column chromatography (CC) over silica gel (PE/EtOAc, *v*/*v*, 100:0→0:100 then EtOAc/MeOH *v*/*v*, 100:0→0:100) to afford six fractions (Fr.1–Fr.6). Fr.1 was separated with Sephadex LH-20 in MeOH to give Fr.1.1–Fr.1.3. Of these, Fr.1.1 was further purified by semi-preparative reverse phase (SP-RP) HPLC with CH_3_CN/H_2_O (*v*/*v* 35:65, 2.0 mL/min) to yield compounds **1** (*t*_R_ = 15.0 min, 8 mg), **3** (*t*_R_ = 25.0 min, 1.1 mg). Compound **1** was able to be crystalized in a mixed organic solvents system (MeOH/DCM/H_2_O, 40:20:1) to afford colorless prisms. Fr.1.2 was subjected to reversed-phase medium-pressure liquid chromatography (MPLC) with MeOH/H_2_O (*v*/*v* 3:7 to 9:1) to offer five sub-fractions (Fr.1.2.1~Fr.1.2.5). Fr.1.2.2 was purified by SP-RP HPLC with CH_3_CN/H_2_O (*v/v* 33:67, 2.0 mL/min) to yield compounds **6** (*t*_R_ = 30.0 min, 0.8 mg), **4** (*t*_R_ = 45.0 min, 1.1 mg). Subfraction Fr.1.2.3 afforded compounds **2** (*t*_R_ = 30.0 min, 10.3 mg) and **5** (*t*_R_ = 78.0 min, 1.7 mg) after purification by SP-RP HPLC with CH_3_CN/H_2_O (*v*/*v* 40:60, 2.0 mL/min).

Compound **1**: Colorless prisms, UV (MeOH) *λ*_max_ (log *ε*) = 208 (4.86) nm, 262 (3.95) nm, 333(3.32) nm, [α]^20^d-63 (*c* = 0.1, MeOH). For ^1^H and ^13^C NMR spectroscopic data, see Table 1; HR-ESI-MS *m/z* 327.0481 [M + Na]^+^ (calcd for C_15_H_12_O_7_Na, 327.0475)

Compound **2**: Colorless prisms, UV (MeOH) *λ*_max_ (log *ε*) = 204 (3.53) nm, 227 (3.23) nm, 275 (3.31) nm, {[α]^25^d-160.0 (*c* = 0.1, MeOH). For ^1^H and ^13^C NMR spectroscopic data, see Table 1; HR-ESI-MS *m/z* 369.0587 [M + Na]^+^ (calcd. for C_17_H_14_O_8_Na, 369.0581).

### 3.5. Single-Crystal X-ray Diffraction Analysis

The crystals obtained for **1** and **2** were tested on a Bruker APEX-II CCD diffractometer through Ga Kα (λ = 1.34139 Å). The structures were solved by direct methods (SHELXT-2014) and refined via full-matrix least-squares difference Fourier techniques using SHELXL-2018/3. Crystallographic data for the structures have been deposited with the Cambridge Crystallographic Data Centre. Copies of the data can be obtained, free of charge, on application to the Director, CCDC, 12 Union Road, Cambridge CB2 1EZ, UK (fax: +44-(0)1223-336033 or e-mail: deposit@ccdc.cam.ac.uk).

Crystallographic data for **1**: C_15_H_12_O_7_, *M*r = 304.25, prism from MeOH/DCM/H_2_O (40:20:1), space group *P*_21_, *a* = 7.7645(5) Å, *b* = 6.7905(5) Å, *c* = 12.6077(9) Å, *V* = 663.16(8) Å3, *Z* = 2, *µ* = 0.669 mm^−1^, F(000) = 316.0; crystal size: 0.200 × 0.180 × 0.170 mm^3^; 2369 unique reflections with 2300 obeying the *I > 2σ(I)*; *R* = 0.0288(2300), *wR*2 = 0.0781(2369), *S* = 1.060; supplementary publication no. CCDC-2027079.

Crystallographic data for **2**: C_17_H_14_O_8_, *M*r = 346.28, prism from MeOH/H_2_O (40:1), space group *P*_-1_, *a* = 7.6247(9) Å, *b* = 8.3365(10) Å, *c* = 14.4259(17) Å, *V* = 881.17(18) Å3, *Z* = 2, *µ* = 0.573 mm^−1^, F(000) = 360.0; crystal size: 0.220 × 0.160 × 0.190 mm^3^; 3092 unique reflections with 1293 obeying the *I > 2σ(I)*; *R* = 0.0709(1293), *wR*2 = 0.1679(3092), *S* = 1.011; supplementary publication no. CCDC-2051826.

### 3.6. Cytotoxicity Assay

CCRF-CEM human T lymphoblast cells were obtained from the National Collection of Authenticated Cell Cultures, Shanghai. Compounds **1**–**5** were diluted serially from 10 uM mother solution in DMSO to make the final concentration from 0 to 20 μM and the final DMSO concentration was ≤0.1% in the reaction mixture. CCRF-CEM cells (5 × 10^4^ in 100 μL) and the tested compounds were co-incubated in a 96-well plate and the absorbance at 490 nm were recorded after published protocols [38,39]. The IC_50_ values were calculated using GraphPad Prism 7 software.

## 4. Conclusions

The chemical investigation of the MCE sponge-associated fungus *Pleosporales* sp. NBUF144, after prioritization by a combination of the OSMAC approach and LC-MS/MS molecular networking, led to the isolation of two new polyketide natural products (**1**–**2**). Compounds **1**–**5** were screened for in vitro cytotoxicity towards CCRF-CEM cells. Compound **1** showed strong cytotoxic activity with an IC_50_ value of 0.46 μM, while **3** was not active at concentrations up to 20 μM, indicating the importance of the Δ3,4 olefin moiety in the scaffold of these compounds for this biological activity. Since compound **4** was also not active, in comparison with **1**, it is suggested that the C-1 and C-2 hydroxy groups are also important for cytotoxicity. Together, this extends the known potential of this chemical class, as drugs leading to the development of new anticancer agents, as well as restricting the flexibility for modification according to the preliminary natural structure-activity relationship (SAR). Further natural product research of sponge-derived fungi, especially those obtained from understudied MCEs, is expected to continue to yield new chemistry and associated biological activity for evaluation in drug development and ecological research.

## Figures and Tables

**Figure 1 marinedrugs-19-00186-f001:**
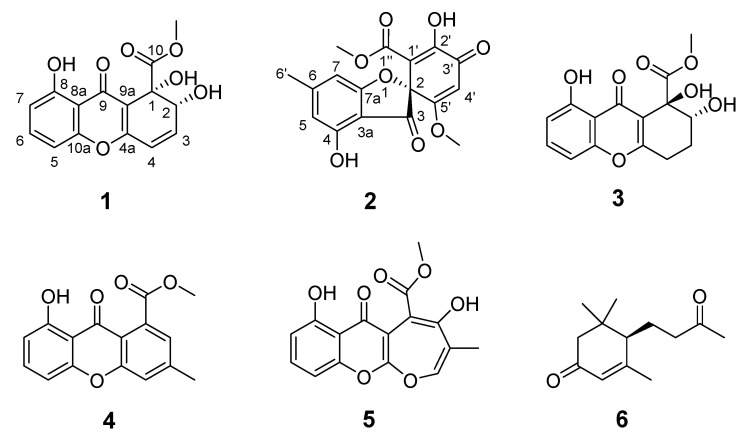
Structures of compounds **1**–**6**.

**Figure 2 marinedrugs-19-00186-f002:**
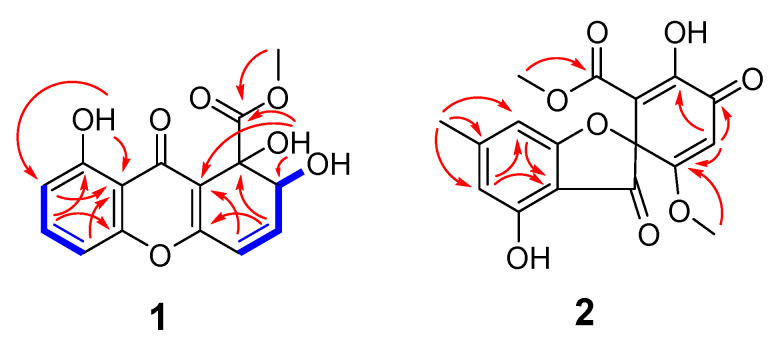
Selected correlations used to determine the planar structures of compounds **1**–**2**. Red single-sided arrows represent cross-peaks from the ^1^H-^13^C HMBC spectrum. Blue double-sided arrows show protons correlated in the ^1^H-^1^H COSY spectrum.

**Figure 3 marinedrugs-19-00186-f003:**
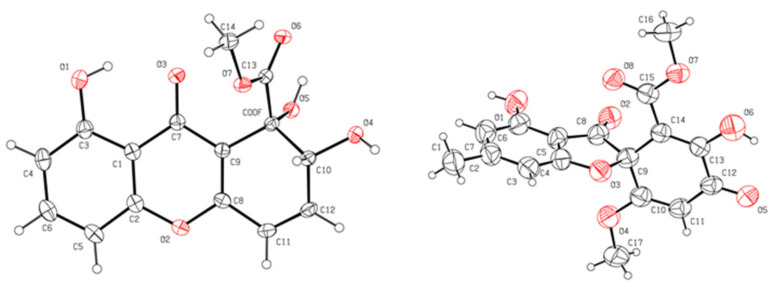
X-ray ORTEP drawings of compounds **1** (**left**) and **2** (**right**).

**Figure 4 marinedrugs-19-00186-f004:**
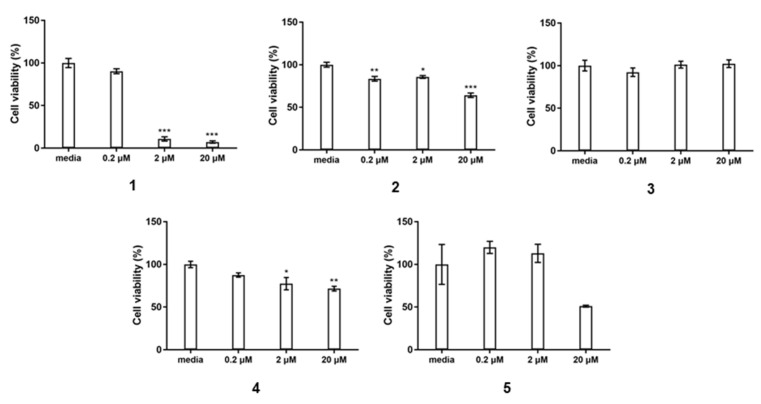
Cell viability of CCRF-CEM cells treated with compounds **1**–**5**. The indicated cell lines were treated for 24 h with tested compounds at gradient concentrations 0.2, 2.0, 20.0 μM, and medium (control). Bars on the bar graphs represent the mean ± SEM versus the control group, * *p* < 0.5, ** *p* < 0.1, *** *p* < 0.01.

**Table 1 marinedrugs-19-00186-t001:** ^1^H and ^13^C NMR spectroscopic data of **1** and **2**(600, 150 MHz, CDCl_3_).

Pos.	1		Pos.	2	
	*δ*_C_, mult	*δ*_H_, Mult (*J* in Hz)		*δ*_C_, mult	*δ*_H_, Mult (*J* in Hz)
1	73.2, C		1		
2	71.5, CH	4.90 d (9.7)	2	93.4, C	
3	143.8, CH	6.50 d (9.8)	3	198.1, C	
4	120.5, CH	6.32 d (9.8)	3a	108.2, C	
4a	160.9, C		4	171.8, C	
5	107.3, CH	6.90 d (8.3)	5	104.6, CH	6.40, s
6	135.8, CH	7.52 t (8.3)	6	152.7, C	
7	112.1, CH	6.80 d (8.2)	7	109.0, CH	6.35, s
8	160.8, C		7a	155.8, C	
8a	111.0, C		1′	149.6, C	
9	180.7, C		2′	148.3, C	
9a	113.7, C		3′	180.9, C	
10	174.6, C		4′	102.5, CH	5.82, s
10a	155.5, C		5′	171.2, C	
1-OH		4.42 s	6′	23.2, CH_3_	2.39, s
2-OH		2.93 s	1″	167.7, C	
8-OH		12.13 s	1″-O*CH*_3_	51.4, CH_3_	3.44, s
10-O*CH*_3_	54.4, CH_3_	3.92 s	5′-O*CH*_3_	57.1, CH_3_	3.69, s

## Data Availability

The datasets generated for this study can be found in the Cambridge Structural Database https://www.ccdc.cam.ac.uk/structures.

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
