# Peer review of "Cytotoxic Polyketide Metabolites from a Marine Mesophotic Zone Chalinidae Sponge-Associated Fungus Pleosporales sp. NBUF144"

_marinedrugs, 2021, doi:10.3390/md19040186_

Round 1

Reviewer 1 Report

This paper summarizes 3 new derivatives of polyketide natural products isolated from the fungus Pleosporales sp. derived from a Chalinidae family sponge, together with their cytotoxicity towards CCRF-CEM human acute lymphatic leukemia cells.

The quality of the paper may be improved if suggested and implements are undertaken. Here are suggestions for a revision.

  1. In the abstract, the sentence “that was derived from a 62 m Chalinidae family deep sponge” may be changed to “that was derived from a 62 m deep Chalinidae family sponge”.
  2. Page 2: Regarding the structure elucidation of compound 1, it was stated as “However, the optical rotation of 1 {[α] 25 D +20.4 (c = 0.1), in MeOH} differs in both sign and magnitude from the reported value for globosuxanthone A {[α] 25 D -50 (c = 0.1), in MeOH/CHCl3 5:3} [28]. This suggested an alternate configuration of C-1 in 1 compared to globosuxanthone A”. This statement is not correct because the difference of optical rotation in sign may indicate the difference in the configuration of C-1, or C-2, or both. The absolute configuration may be deduced from X-ray crystallography. And the word “alternate” should be changed to “alternative”.
  3. Page 3 and 4: The determinations of the absolute configurations of compound 2 is not very clear. The absolute configuration of compound 2 was assigned to be the same as that of compound 1, because they are produced by the same organism. However, their quite different optical rotation data (+ 20 / -86) make me hesitate to accept that they share the same absolute stereochemistry. Further extensive analysis and discussion would be necessary. The configuration of C-2 should be also considered. Mosher method may be applied to determine the absolute configuration of hydroxyl groups.
  4. ECD data was utilized to provide additional support to the absolute configuration of compound 2. However, it would be better to compare calculations of all 4 stereoisomers.
  5. The absolute configuration of compound 4 was assigned to be the same as that of globosuxanthone A, because the sign of the optical rotation of compound 4 was opposite to that of compound 2. However, it cannot be easily accepted that compounds 4 and globosuxanthone A, which have the same absolute configuration, would show quite different optical rotation data (+196 / -50). Again further extensive analysis and discussion would be necessary. The configuration of C-2 should be also considered.
  6. Page 6: The numbering of compound 3 in the main text dose not match with those of Figure 1 and Table 3. The numbering (4’) in the main text may be corrected.

Author Response

Dear Sir/Madam,

  Thank you for your valuable comments. We are happy to revise our manuscript according to your suggestions. The item to item responses in dark blue font are listed below. Supplementary Figures and Tables are attached.

Comments and suggestions for authors from reviewer 1: This paper summarizes 3 new derivatives of polyketide natural products isolated from the fungus Pleosporales sp. derived from a Chalinidae family sponge, together with their cytotoxicity towards CCRF-CEM human acute lymphatic leukemia cells. The quality of the paper may be improved if suggested and implements are undertaken. Here are suggestions for a revision.

Question 1: In the abstract, the sentence “that was derived from a 62 m Chalinidae family deep sponge” may be changed to “that was derived from a 62 m deep Chalinidae family sponge”

Answer: Thank you for suggestion. We have corrected “a 62 m Chalinidae family deep sponge” into “a 62 m deep Chalinidae family sponge”.

Question 2: Page 2: Regarding the structure elucidation of compound 1, it was stated as “However, the optical rotation of 1 {[α]25 D +20.4 (c = 0.1), in MeOH} differs in both sign and magnitude from the reported value for globosuxanthone A {[α]25 D -50 (c = 0.1), in MeOH/CHCl3 5:3} [28]. This suggested an alternate configuration of C-1 in 1 compared to globosuxanthone A”. This statement is not correct because the difference of optical rotation in sign may indicate the difference in the configuration of C-1, or C-2, or both. The absolute configuration may be deduced from X-ray crystallography. And the word “alternate” should be changed to “alternative”.

Answer: Thank you for reminding. We have changed “an alternate configuration of C-1 in 1 compared to globosuxanthone A” into “This suggested an alternative stereochemistry of 1 compared to globosuxanthone A”.

Question 3: Page 3 and 4: The determinations of the absolute configurations of compound 2 is not very clear. The absolute configuration of compound 2 was assigned to be the same as that of compound 1, because they are produced by the same organism. However, their quite different optical rotation data (+ 20 / -86) make me hesitate to accept that they share the same absolute stereochemistry. Further extensive analysis and discussion would be necessary. The configuration of C-2 should be also considered. Mosher method may be applied to determine the absolute configuration of hydroxyl groups.

Answer: We’re grateful for your suggestions. We have re-tested the optical rotation (OR) values of compounds 1 and 2 (original numbering system), but get different results {[α]20 D-63 (c = 0.1, MeOH), 1; [α]20 D+39 (c = 0.08, MeOH), 2). There maybe two reasons behind this phenomenon: The key step of zeroing the blank solvent (MeOH) when one of the co-authors collected the OR values; the polarimeter lacks routine maintenance and didn’t work steadily. The updated OR data are convincing for we have tested the values of D-glucose and compared the results with literatures to make sure the polarimeter works well.

However, the re-tested OR values of compounds 1 and 2 (original numbering system) still have opposite signs. We have checked the literatures and found that the OR values of known compounds globosuxanthone A {[α]25 D-50 (c = 0.1), in MeOH/CHCl3 5:3} and 3,4-dihydroglobosuxanthone A{[α]20 D +38 (c = 0.07, CDCl3)} also have opposite signs [1,2]. The reason for this maybe the absent double bond in 3,4-dihydroglobosuxanthone A or compound 2 in A rings (as shown in the figure below) forms a chair conformation which impact more on their OR values.

We totally agree with you that Mosher method may be the right way to determine the absolute configuration of hydroxyl groups. But the quantity of the remaining compound 2 is less than 0.5 mg, it’s too difficult for us to do Mosher with the sample.

Question 4: ECD data was utilized to provide additional support to the absolute configuration of compound 2. However, it would be better to compare calculations of all 4 stereoisomers.

Answer: Thank you for your suggestions. The NMR data of compounds 2 and 4 (original numbering system) in this study suggested that they share planar structures. There are 4 stereoisomers should be considered, which process (1S,2R), (1R,2S), (1R,2R) and (1S,2S) configurations. It can be deduced that 1H and 13C NMR chemical shifts of compound 2 (original numbering system) could identical to those of compound 4 (original numbering system) if they process the same (1R*,2R*) configuration. But in the same NMR solvent the chemical shift differences [especially at C-1 (ΔδC 3.3 ppm) and C-2 (ΔδC 2.2 ppm)] indicate that these molecules must be diastereomeric. As reported, DP4 method and improved DP4+ were well established for stereochemical study [3-7]. Thus, we decided to employ DP4+ to assign relative configurations of compounds 2 and 4 (original numbering system) before the absolute stereochemistry determination. DP4+ indicating the (1R*,2R*)-configuration (DP4+ = 100%) of compound 4 (original numbering system), together with the comparable optical rotations {[α]20 D +19 (c = 0.08, MeOH), 4 (original numbering system); [α]20 D (original numbering system)  +38 (c = 0.07, CDCl3), 3,4-dihydroglobosuxanthone A}, compound 4 (original numbering system)  was determined to share (1R,2R)-configuration with 3,4-dihydroglobosuxanthone A.

However, we didn’t get enough satisfied NMR calculation data for compound 2. The (1S*,2R*)-configuration of compound 2 (DP4+ = 66.7%) doesn’t seem to have absolute advantage against the (1S*,2R*)-configuration of compound 2 (DP4+ = 33.3%). Nonetheless, we still have tried ECD calculation with GIAO method at mPW1PW91-SCRF/6-31+G(d,p) level with IEFPCM solvent model different with the one described in the first version of manuscript to solve the configuration of compound 2 (original numbering system). Based on the ECD calculation results, compound 2 seems to have (1S,2R) configuration.

In sum, chemical calculations indicated the (1S,2R) configuration of 2, but not solid enough. Different OR values from different data collection often brought confusion in this problem. According to your suggestions, Mosher method or hydrogenation reaction from 1 to 2 is the most reasonable way to solve this problem. However, our remaining sample really can not support these chemical derivation reactions. To avoid publishing scientific mistakes, we decided not to report compound 2 (original numbering system) this time. We chose to cultivate this fungal strain again to enrich this compound, so we can determine its configuration with X-ray analysis or chemical reactions.

Reference:

[1]  Hussain, H.; Krohn, K.; Floerke, U.; Schulz, B.; Draeger, S.; Pescitelli, G.; Antus, S.; Kurtán, T. Absolute configurations of globosuxanthone A and secondary metabolites from Microdiplodia sp. – a novel solid-state CD/TDDFT approach. Eur. J. Org. Chem. 2007, 292−295.

[2]  Krohn, K.; Kouam, S.F.; Kuigoua, G.M.; Hussain, H.; Cludius-Brandt, S.; Flörke, U.; Kurtán, T.; Pescitelli, G.; Di Bari, L.; Draeger, S.; Schulz, B. Xanthones and oxepino[2, 3-b]chromones from three endophytic fungi. Chem. Eur. J. 2010, 15, 12121–12132.

[3]  Willoughby, P.H.; Jansma, M.J.; Hoye, T.R. A guide to small-molecule structure assignment through computation of (1H and 13C) NMR chemical shifts. Nat. Prot. 2014, 9, 643−660.

[4]  Grimblat, N.; Zanardi, M.M.; Sarotti, A.M. Beyond DP4: an improved probability for the stereochemical assignment of isomeric compounds using quantum chemical calculations of NMR shifts. J. org. chem. 2015, 1−24.

[5]  Semenov, V.A.; Krivdin, L.B. DFT computational schemes for 1H and 13C NMR chemical shifts of natural products, exemplified by strychnine. Magn. Reson. Chem. 2019, 58, 1−9.

[6]  Semenov, V.A.; Krivdin, L.B. Computational 1H and 13C NMR of the trimeric monoterpenoid indole alkaloid strychnohexamine: selected spectral updates. Magn. Reson. Chem. 2021, 1−10.

[7]  Zanardi, M.M. Suárez, A.G.; Sarotti, A.M. Determination of the relative configuration of terminal and spiroepoxides by computational methods. Advantages of the Inclusion of Unscaled Data. J. Org. Chem. 2017, 82, 1873−1879.

Question 5: The absolute configuration of compound 4 was assigned to be the same as that of globosuxanthone A, because the sign of the optical rotation of compound 4 was opposite to that of compound 2. However, it cannot be easily accepted that compounds 4 and globosuxanthone A, which have the same absolute configuration, would show quite different optical rotation data (+196 / -50). Again further extensive analysis and discussion would be necessary. The configuration of C-2 should be also considered.

Answer: Thank you for your suggestion. We have re-tested the optical rotations of compounds 1, 2 and 4 {[α]20 D+39 (c = 0.08, MeOH)} (original numbering system). We have made detailed discussion about the OR values of globosuxanthone A and 3,4-dihydroglobosuxanthone A in the answer to question 3. In addition, we have added NMR calculations of 4 to assign its relative configuration and more discussion about the absolute stereochemistry of 2.

Question 6: Page 6: The numbering of compound 3 in the main text dose not match with those of Figure 1 and Table 3. The numbering (4’) in the main text may be corrected.

Answer: Thank you for your reminding. We have corrected (4’) into (2’) in the relevant part in the main text, and the name of compound 3 (now renumbered as compound 2) 4’-hydroxy bisdechlorogeodin into 2’-hydroxy bisdechlorogeodin.

Thanks for your attention and patience, and look forward to hearing from you.

Sincerely yours,

Tingting Wang

[email protected]

Li Dak Sum Marine Biopharmaceutical Research Center, Department of Marine Pharmacy, College of Food and Pharmaceutical Sciences, Ningbo University, Ningbo, Zhejiang, 315800, China

Reviewer 2 Report

This is a very nice work. It is well executed, argued and articulated.

There is a good introduction and setting of the context. The methods, results and conclusions are similarly in good form. I do have some small suggestions and exceptions:

Figure 3 the ORTEP drawings (figure 3) miss the point of ORTEP ellipsoids: to convey the uncertainty of the atomic positions or to convey the disorder of them. The ellipsoids are so small as to convey nothing (to my eye). Line 164 states the planarity of 3 is depicted in figure 3. This can be performed without ORTEP ellipsoids and perhaps with a second complimentary orientation of either or both of the molecules.

This is a stylistic suggestion (only). If I were creating Figure 5 I would use "[1] (uM)" and "[2] (uM)" and so on for the x axis labeling which conveys the same information, in a more compact form and emphasizes the standard chemical terminology. I would also leave out the "CCRF-CEM" excel-like title off of the figure.

In line 205 the chemical shift values must be referenced, such as "The chemical shift values are given in parts per million (ppm) relative to TMS at 0.0 ppm."

Lines 222 and 229 refer to "potato" and "potato dextrose", respectively. is this correct?

Lines 230-232 are a bit confusing as written. Which of the filtrate or the solids retained in the cheese cloth (the mycelia) is subsequently extracted.

I think I understand what lines 233-234 mean - the TLC analyses of each of the solvent extracts were similar so they were combined. If I'm correct or not this might be better written.

I do not know what "... total crude organic extract (348.7g, partially wet weight)" means in a number of aspects. First of all, the number of significant figures is ridiculously to large. 350 g would be sufficiently precise and demonstrate better analytic chemistry form. Second, does this refer to the solids after removal of the solvent? If so, then it ought to be stated. If not and it refers to the volume of solution (as a mass) then it is superflurious and ought not to be included. The mass of solids ought to be stated. Third, what does "partially wet weight" mean? Either it is wet or is is dried (according to specified conditions, for example, 40 deg C and 10^-2 Torr for one hour).

In lines 278-279 I'm left wondering exactly how the solutions were created. If the compounds were diluted in DMSO how did the final [DMSO] reach such a low value? I often see beginning analytic chemists include "serially" in their descriptions where it adds no information of assistance. I see no rationale for its inclusion here. Also, "the OD values on 490 nm" on line 281 is better written as "the absorbance at 490 nm"

Author Response

Dear Sir/Madam,

  Thank you for your valuable comments. We are happy to revise our manuscript according to your suggestions. The item to item responses in dark blue font are listed below.  Supplementary Figures and Tables are attached.

Comments and suggestions for authors from reviewer 2: This is a very nice work. It is well executed, argued and articulated. There is a good introduction and setting of the context. The methods, results and conclusions are similarly in good form. I do have some small suggestions and exceptions:

Question 1: Figure 3 the ORTEP drawings (figure 3) miss the point of ORTEP ellipsoids: to convey the uncertainty of the atomic positions or to convey the disorder of them. The ellipsoids are so small as to convey nothing (to my eye). Line 164 states the planarity of 3 is depicted in figure 3. This can be performed without ORTEP ellipsoids and perhaps with a second complimentary orientation of either or both of the molecules.

Answer: Thank you for your suggestion. We have renewed the X-ray ORTEP drawings of compounds 1 and 3 (Now renumbered as 2). Because the X-ray structure of compound 2 plays no role in determining its absolute configuration, we have made some changes in the text to explain this. We have changed “Single-crystal X-ray diffraction analysis was used to confirm the planar structure of 2 (Figure 3), but it could not establish the C-2 configuration due to the centrosymmetric space group P-1 of the crystals obtained.” into “Single-crystal X-ray diffraction analysis was attempted to establish the C-2 configuration of 2 (Figure 3), but since a centrosymmetric space group (P-1) was obtained with the crystals, only confirmation of the planar structure of 2 was achieved.”

Question 2: This is a stylistic suggestion (only). If I were creating Figure 5 I would use "[1] (uM)" and "[2] (uM)" and so on for the x axis labeling which conveys the same information, in a more compact form and emphasizes the standard chemical terminology. I would also leave out the "CCRF-CEM" excel-like title off of the figure.

Answer: Thank you. We have made some changes in Figure 5 (now renumbered as Figure 4) according to your valuable suggestions. 

Question 3: In line 205 the chemical shift values must be referenced, such as "The chemical shift values are given in parts per million (ppm) relative to TMS at 0.0 ppm."

Answer: Thank you for your kind advice. We have added “relative to TMS at 0.0 ppm” in the “3.1. General” section.

Question 4: Lines 222 and 229 refer to "potato" and "potato dextrose", respectively. Is this correct?

Answer: Thank you for your reminding. “Potato” is the correct constituent of the media, and we have delete “dextrose” in the text.

Question 5: Lines 230-232 are a bit confusing as written. Which of the filtrate or the solids retained in the cheese cloth (the mycelia) is subsequently extracted.

Answer: Thank you for reminding. We have made some changes in the text to detail the extraction work.

We have changed “The culture broth was filtered through cheese cloth to separate it into filtrate and mycelia, which were extracted with…” to “The culture broth was filtered through cheese cloth to separate it into filtrate and mycelia (retained in the cheese), and both of them were extracted with…”.

We also have changed “The obtained extracts were combined for their similar spots in TLC analysis to afford a total crude organic extract” to “The separated extracts of the filtrate and mycelia were combined for their similar spots in TLC analysis to afford a total crude organic extract”. We have added “filtrate and mycelia” behind “extracts” to explain the source of extracts.

Question 6: I think I understand what lines 233-234 mean - the TLC analyses of each of the solvent extracts were similar so they were combined. If I'm correct or not this might be better written.

Answer: Thank you. Improved as requested (see above).

Question 7: I do not know what "... total crude organic extract (348.7g, partially wet weight)" means in a number of aspects. First of all, the number of significant figures is ridiculously to large. 350 g would be sufficiently precise and demonstrate better analytic chemistry form. Second, does this refer to the solids after removal of the solvent? If so, then it ought to be stated. If not and it refers to the volume of solution (as a mass) then it is superflurious and ought not to be included. The mass of solids ought to be stated. Third, what does "partially wet weight" mean? Either it is wet or is is dried (according to specified conditions, for example, 40 deg C and 10^-2 Torr for one hour).

Answer: Thank you for your reminding. We have checked the test records and corrected the total weight of “348.7 g (partially wet weight)” into dry weight “280 g”.

Question 8: In lines 278-279 I'm left wondering exactly how the solutions were created. If the compounds were diluted in DMSO how did the final [DMSO] reach such a low value? I often see beginning analytic chemists include "serially" in their descriptions where it adds no information of assistance. I see no rationale for its inclusion here. Also, "the OD values on 490 nm" on line 281 is better written as "the absorbance at 490 nm".

Answer: Thank you for your suggestions. We created solutions at the final concentration (0 to 20 μM) from mother solution with a concentration of 10 mM by serial dilution, to make sure the final DMSO reaching a low value.

We have corrected “the OD values on 490 nm” into “the absorbance at 490 nm” according to your suggestion.

Thanks for your attention and patience, and look forward to hearing from you.

Sincerely yours,

Tingting Wang

[email protected]

Li Dak Sum Marine Biopharmaceutical Research Center, Department of Marine Pharmacy, College of Food and Pharmaceutical Sciences, Ningbo University, Ningbo, Zhejiang, 315800, China

Reviewer 3 Report

I think this would be a good article in Molecules. The only change I recommend that one less significant digit be used on the reported optical rotation data.

Author Response

Dear Sir/Madam,

  Thank you for your valuable comments. We are happy to revise our manuscript according to your suggestions. The item to item responses in dark blue font are listed below.

Comments and suggestions for authors from reviewer 3: I think this would be a good article in Molecules.

Question: The only change I recommend that one less significant digit be used on the reported optical rotation data.

Answer: Thank you. We have used one less significant digit on the optical rotation data according to your suggestion.

Thanks for your attention and patience, and look forward to hearing from you.

Sincerely yours,

Tingting Wang

[email protected]

Li Dak Sum Marine Biopharmaceutical Research Center, Department of Marine Pharmacy, College of Food and Pharmaceutical Sciences, Ningbo University, Ningbo, Zhejiang, 315800, China

Round 2

Reviewer 1 Report

A substantial revision was made to the manuscript, and I recommend this manuscript for publication to Mar. Drugs